# Case-control study on determinants of uterine rupture among mothers who gave birth at Hawassa University comprehensive specialized hospital

**Getnet Feleke**[1◉], **Temesgen Tantu**[2◉]*, **Dereje Zewdu**[3‡], **Abel Gedefawu**[4◉], **Mekete Wondosen**[5‡], **Muluken Gunta**[6‡]

1 Obstetrics and Gynecology in Arbaminch University College of Medicine and Health Sciences, Arbaminch, Ethiopia, 2 Obstetrics and Gynecology in Wolkite University College of Medicine and Health Sciences, Wolkite, Ethiopia, 3 Anesthesia in Wolkite University College of Medicine and Health Sciences, Wolkite, Ethiopia, 4 Obstetrics and Gynecology in Hawassa University College of Medicine and Health Sciences, Hawassa, Ethiopia, 5 Surgery in Wolkite University College of Medicine and Health Sciences, Wolkite, Ethiopia, 6 MPH, Wolaita Zone Health Department, Wolaita Sodo, Ethiopia

◉ These authors contributed equally to this work.
‡ DZ, MW and MG also contributed equally to this work.
* tematantu405@gmail.com

**Data Availability Statement:** All relevant data are within the paper and its Supporting Information files.

## Abstract

### Background

Uterine rupture is defined as tearing of the uterine wall during pregnancy or delivery. It can occur during pregnancy or labor and delivery. Rupture of the uterus is a catastrophic event resulting in the death of the baby, and severe maternal morbidity and mortality Despite different interventions done by stakeholders, it remained one of the leading public problems in developing countries like Ethiopia.

### Objective

This study assessed the prevalence and determinants of uterine rupture among mothers who gave birth at Hawassa University comprehensive specialized hospital from July 2015 to June 2020G.C.

### Method

A case-control study was conducted by reviewing data from a total of 582 patient charts which include 194 cases and 388 controls with a case-to-control ratio of 1:2. Then the data was extracted using a pre-tested and structured data extraction sheet. Data were entered using Epi data 3.1 and exported to SPSS and analyzed using SPSS 20. The association between independent variables and uterine rupture was estimated using an odds ratio with 95% confidence intervals. The statistical significance of the association was declared at P-value < 0.05.

**Funding:** The author(s) received no specific funding for this work.

**Competing interests:** The authors have declared that no competing interests exist.

## Result

There were a total of 22,586 deliveries and 247 confirmed cases of uterine rupture which makes the prevalence 1.09%. Lack of ANC (Ante-natal care) (AOR = 7.5; 95% CI: 1.9–30.3) inadequate ANC (AOR = 2.45; 95% CI: 1.1–5.57), gravidity ≥5 (AOR = 3.3; 95% CI: 1.36–8.12), obstructed labor (AOR = 38.3; 95% CI: 17.8–82.4) and fetal macrosomia (AOR = 8; 95% CI: 17.8–82.4) are variables which increase the odds of developing uterine rupture. Mothers without additional medical or obstetric conditions are more likely (AOR = 4.2; 95% CI: 2.1–8.65) to develop uterine rupture than mothers with additional medical or obstetric conditions.

## Conclusion

The prevalence of uterine rupture is high in the study area. The study also revealed that a decrease in ANC follow-up, gravidity of ≥5, obstructed labor, and fetal weight of >4kg are significantly associated with uterine rupture. Improving the quality of ANC follow-up, intra-partum follow-up and proper estimation of fetal weight are recommended interventions from the study.

## Background

Pregnancy is supposed to be a state of happiness and well beingness but sometimes it will end up with multiple catastrophic complications costing maternal life. One of these is uterine rupture which is known as the tearing apart of the uterine wall. It can be complete or incomplete [1]. Complete uterine rupture is when the tearing involves the whole layers of the uterine wall including the serosa. An incomplete uterine rupture involves mucosa and myometrium but the serosa is not involved. Uterine rupture can occur during pregnancy, during labor without dystocia, or following obstructed labor [2]. The first two are common in developed countries since they are usually associated with previously scarred uterus whereas the last one is common in developing countries where the prevalence of labor abnormalities is higher [1–5].

The prevalence of uterine rupture reported was considerably lower for population-based (median 0.053%, range 0.016–0.30%) than for facility-based studies (median 0.31%, range 0.012–2.9%) [1]. The prevalence range between 0.006% for women without a previous cesarean section in a developed country and 25% for women with obstructed labor in the least developed country. The overall prevalence of uterine rupture in those who had uterine scar is 0.5% [3]. In highly developed countries the prevalence is 0.2% whereas in the least developed countries prevalence of rupture after the uterine scar is 1% [3, 6, 7]. Studies done in developing countries showed a high prevalence of uterine rupture ranging from 0.12% - 3.38 [4, 7–9]. Based on facility base studies conducted in Ethiopia the prevalence ranges from 0.9% - 16.68% [10–14].

Globally maternal mortality is unacceptably high and 94% of these deaths occurred in low-resource settings, of these Sub-Saharan Africa alone accounted for roughly two-thirds of maternal deaths in 2017 [6]. According to a systematic analysis obstructed labor accounts for 22.34% of maternal death in Ethiopia, which is the second most common cause of maternal death next to hemorrhage [15]. Maternal death secondary to uterine rupture ranges from 2.1 to 11.2% [10–14]. Mothers who survived also end up with other obstetric complications

including an obstetric fistula, anemia, sepsis, and fetal loss [10–14]. But the fetal loss in rupture of the scarred uterus is lower as compared to rupture of the unscarred uterus [1, 3].

Many risk factors are associated with an increased risk of uterine rupture. Among these high parities, lack of ANC, rural residency, malpresentation, previous history of uterine scar, congenital anomalies, lack of using partograph, obstructed labor, and macrosomia are some of the risk factors associated with uterine rupture [10, 16–23].

Though there are some studies done in a different part of the country, it is not adequate and showed discrepancies in factors associated with the problem. Hence reducing the maternal mortality rate is one of the goals of sustainable development goal (SDG). The global goal is to reduce the maternal mortality ratio to less than 70 per 100,000 deliveries by 2030 G.C [22]. Ethiopia is far from this target. According to EDHS (Ethiopian Demographic health survey) 2016, MMR is 412 per 100,000 deliveries [24]. Studies like this one can help to identify the burden and determinants of uterine rupture.

## Method and material

### Study area and study design

An institution-based unmatched case-control study was conducted from May 2020 –June 30/2020 G.C in Hawassa University comprehensive specialized hospital (HUCSH) located in the southern part of Ethiopia and 278km from Addis Ababa; capital of Ethiopia. Hawassa city is home to 315,267 people. HUSCH is a tertiary hospital serving the town and surrounding area. It serves more than 12 million people. The hospital is also a teaching hospital currently running 40 OBGYN residents and undergraduate medical students.

### Sampling participants

All mothers who gave birth at Hawassa University comprehensive specialized Hospital are the source population of the study. Mothers who were diagnosed to have uterine rupture and managed for Uterine rupture were considered as cases whereas those mothers who gave birth through vaginal deliveries were taken as controls. The sample size was determined using the double population proportion formula for case-control study design using Epi Info version 3.1 statistical software with consideration of the following assumptions: Power of the study = 80%, Confidence interval = 95%, Case–to- control ratio = 1 to 2, the proportion of case with exposure 13.6% [13] to have calculated sample size 588 (194 cases and 388 controls). After excluding those cards which were not eligible, only 200 cards left then we took the 194 cards as cases. The data collected from on the client's medical registration number recorded on the log books in the labor wards and operating rooms over the 5 years from July 2015 to June 2020. But to calculate the prevalence, all mothers who develop uterine rupture and were managed in Hawassa University comprehensive specialized Hospital from July 2015 to June 2020 were included in the study. Those mothers who gave birth in HUCSH and registered immediately before and after the selected case of uterine rupture was chosen as a control. If the chart is not identified the next mother who fulfills the criteria are used.

**Exclusion criteria.** *Case*. All mothers who were managed for uterine rupture in another Hospital and referred to HUCSH for management of complications of uterine rupture were excluded. Those mothers with incomplete data on the chart were also excluded (those with a lack of more than 20% of data). Those with lost charts were excluded. Those with medico-legal cases were excluded since difficult to retrieve charts.

*Control*. All mothers who gave birth in another hospital and were referred for management of obstetric complications will be excluded. Those with incomplete information and lost charts were also be excluded.

## Data collection tools and procedure

After reviewing different available studies, the data extraction sheet was developed using previous studies and the variables that are going to be studied then it was tested by taking 5% of the sample population at Adare general hospital in Hawassa city. The data were extracted after cross-checking from patient cards, delivery, and operation theater registry. It includes socio-demographic characteristics, pregnancy-related characteristics, and labor and delivery-related characteristics. Data were collected by trained medical interns and junior gynecology and obstetrics residents. After the collection of the data, the completeness of the data extraction sheet was checked by the primary investigator before analysis.

## Data management and analysis

Data extracted were coded and entered in Epidata version 3.1 software for cleaning and then exported to SPSS version 20 for further analysis and report. The descriptive statistical analyses were reported with simple frequencies, crosstabs, and mean and standard deviation to describe sociodemographic characteristics of participants, pregnancy-related and labor and delivery–related characteristics. Binary logistic regression with 95% CI was used to explore the relationship between uterine rupture and the independent variables. Those variables with p values less than 0.25 in bivariate analysis were taken into multivariate analyses for further strengthening of the analysis. Then the variables with a p-value less than 0.05 in multivariate analyses will be declared as having a statistically significant association with uterine rupture. Model fitness was measured using the Hosmer and Lemeshow goodness of fit measures and the Nagelkerke R Square, which were 0.64 and 0.58, respectively. The variance inflation factor (VIF>10) was used to test for multicollinearity between the explanatory variables.

## Result

### Trends of uterine rupture

A total of 582 Patient charts were retrieved (194 cases and 388 control) and data were extracted from the patient chart. Charts with incomplete information, lost charts, and those charts held for the medicolegal issue were not included then it became only 200 charts were eligible. During the five-year study period, from July 2015 to June 2020, there was a total of 22,586 deliveries. Of all, there were 247 confirmed cases of uterine rupture. So that the prevalence of uterine rupture is 1.09% (95% CI: 0.694–1.486). From uterine rupture cases, there were 15 cases of maternal death which makes case fatality 6.07%. The trend of uterine rupture in the five years study period is declining from 2.02% to 0.4% (Fig 1).

### Socio-demographic and pregnancy-related characteristics

The mean age in the case is 29.11 and in control is 26.68 years of age. Most (76.3%) rupture cases and 87.9% of controls are in the age group of fewer than 35 years of age. Most (76.3%) of the cases live in a rural area but most (78.1%) of the controls are from urban areas (Table 1). Most of the cases (84%) and controls (97.2%) had ANC follow-up at least once (Table 2).

### Labor and delivery-related characteristics

Most cases (96.4%) and controls (85.3%) had spontaneous onset of labor whereas five cases (2.6%) and twenty-seven (7%) of controls had induced labor. Two mothers, who had previous cesarean scars sustained uterine rupture, before the onset of labor. Among the cases, 154 (79.4%) had obstructed labor but only 12(3.1%) of the controls had obstructed labor. Obstructed labor is the most common (78.9%) cause of uterine rupture followed by scar

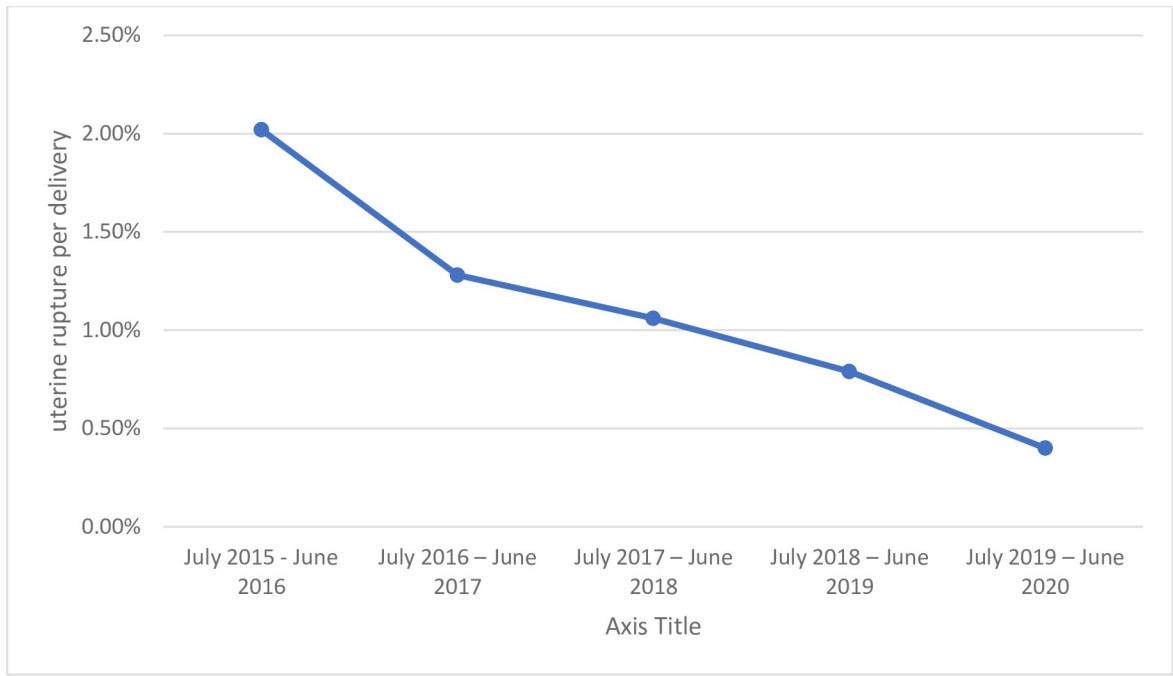

**Fig 1. Trends of uterine rupture of a mother who gave birth at HUCSH from July 2015 to June 2020.**

dehiscence (14.4). Four uterine rupture cases are related to using uterotonic agents and three are related to instrumental delivery. Two mothers experienced uterine ruptures because of fundal pressure given by health professionals during the second stage of labor and four uterine ruptures occur due to precipitated labor (Table 2).

### Health system-related characteristics

More than a quarter (26.3%) of uterine ruptures occur at home and the rest occur either in a health facility or during transport from one facility to another facility. Except for one case of uterine rupture all cases had visited at least one health facility before arrival at HUCSH. The mean duration of hospital stay for cases is 8.08(±6.56) days whereas the mean duration of hospital stay for controls is 2.19(±2.3) days (Table 3).

### Outcome related characteristics

More than half (66%) of rupture cases were in hypovolemic shock during arrival. There were fifteen maternal deaths related to uterine rupture during the study period four of them died before surgery. Total abdominal hysterectomy was done for most (85%) of uterine rupture cases and rupture was repaired for 17(8.8%) cases of uterine rupture. The fetal outcome for uterine rupture cases is very poor; 95.4% of fetuses died before delivery. Nine babies were delivered alive but four of them had a first-minute APGAR score of less than seven (Table 4).

### Determinants of uterine rupture

Those variables which are found to be significant with p-value of less than 0.25 on bivariate logistic regression were exported to multivariate analysis for further association. Having no ANC follow up (AOR = 7.5; 95% CI: 1.9–30.3), Number of ANC follow up (less than four ANC) (AOR = 2.9; 95% CI: (1.3–6.5)), gravidity more than 5 (AOR = 3.4; 95% CI: 1.4–8.3),

**Table 1. Socio-demographic characteristics and pregnancy-related characteristics of the mother who gave birth at HUCSH 2020.**

| Variables | Responses | Case (194) | Control (388) | P value |
|---|---|---|---|---|
| | | Frequency (%) | Frequency (%) | |
| Age(years) | <35 | 148(76.3) | 341(87.9) | 12.958(0.001) |
| | ≥ 35 | 46(23.7) | 47(12.1) | |
| | Mean (SD) | 29.11(±4.82) | 26.68(±4.81) | |
| Residency | Urban | 46(23.7) | 303(78.1) | 159.22 (0.001) |
| | Rural | 148(76.3) | 85(21.9) | |
| Address | Hawassa city | 6(3.1) | 156(40.2) | 0.823 (0.491) |
| | SNNPR | 56(28.9) | 84(21.7) | |
| | Oromia Region | 132(68) | 148(38.1) | |
| Number of ANC visit | No ANC | 31(16) | 11(2.8) | 33.373(0.00) |
| | 1–3 times | 147 (75.8) | 159(41) | |
| | Four and above | 16(8.2) | 218(56.2) | |
| Gravidity | Parity 1–4 | 95(49) | 353(91) | 128.78(0.00) |
| | Parity > = 5 | 99(51) | 35(9) | |
| Gestational age | <37week | 3(1.5) | 17(4.4) | 3.13(0.073) |
| | > = 37week | 191(98.5) | 371(95.6) | |
| Number of fetuses | Singleton | 193(99.5) | 373(96.1) | 5.43(0.02) |
| | Multiple | 1(0.5) | 15(3.9) | |
| Presence of uterine scar | Yes * | 28(14.4) | 67(17.3) | 0.761(0.363) |
| | No | 166(85.6) | 321(82.7) | |
| Duration since the last uterine scar | < 24 months | 0 | 5(7.5) | 13.4(0.145) |
| | > = 24 months | 28(100) | 62(92.5) | |
| Fetal congenital anomalies | Yes Ω | 2(1) | 0 | 4.014(0.045) |
| | No | 192(99) | 388(100) | |
| Other medical/obstetric conditions | Yes | 9(4.6) | 67(17.3) | 20.02 (0.001) |
| | NO | 185(95.4) | 321(82.7) | |

SNNPR = Southern nation, nationality and people's region; SD = Standard deviation.

➢ * = All are cesarean scars.

➢ Ω = Both were hydrocephalus.

➢ Medical/obstetric disorder includes: Antepartum hemorrhage, Hypertension, PROM, cardiac disease, thyrotoxicosis, and renal disease.

➢ SNNPR: southern nation nationality and people's region.

obstructed labor (AOR = 38.3; 95% CI: 17.8–82.4), fetal weight greater than 4000 gram (AOR = 8.4; 95% CI; 3.4–20.8) and other medical or obstetric disorders are variables which showed significant association with uterine rupture (AOR = 5.1; 95% CI 1.6–16.1) (Table 5).

## Discussion

The study showed that the prevalence of uterine rupture is 1.09%. This is comparable with the prevalence of uterine rupture determined by WHO in the least developed countries which was 0.96% [1] and studies done in Ethiopia at Felege Hiwot Hospital [0.9%] [1, 14], Adigrat Hospital (0.9%) [12], and Mizan Aman Hospital [1.24%] [11]. But it is lower than the studies done in the Amhara region referral Hospital [16.68%] [17] and Debre Markos Hospital [2.44%] [10]. On the other hand, it is much higher than the prevalence in developed countries like Belgium [0.036%] [2], Norway [0.5%] [25], the UK [0.02%] [21] and Saud Arabia 7 [0.05%] [26] and some developing countries like Tanzania [0.225%] [4]. These differences with developed countries could be because of the difference in availability of health facilities, referral systems, and infrastructure.

**Table 2. Labor and delivery-related characteristics of a mother who gave birth at HUCSH 2020.**

| Variable | Response | Case (194) | Control (388) | X² (P value) |
|---|---|---|---|---|
| | | Frequency (%) | Frequency (%) | |
| Onset of labor | Spontaneous | 187(97.4) | 331(92.5) | 0.86(0.020) |
| | Induced | 5(2.6) | 27(7.5) | |
| Use of partograph | Yes | 0 | 232(59.8) | 362 (0.001) |
| | No | 31(16) | 129(33.2) | |
| | Unknown | 163(84) | 27⁺(7) | |
| Labor abnormality on partograph | Yes | NA | 17(7.3) | 363(0.001) |
| | No | NA | 215(92.7) | |
| Obstructed labor | Yes | 154(79.4) | 12(3.1) | 369 (0.001) |
| | No | 40(20.6) | 376(96.9) | |
| Trial of instrumental delivery | Yes | 5(2.6) | 6(1.5) | 0.79(0.389) |
| | No | 189(97.4) | 382(98.5) | |
| Use of uterotonics | Yes | 5(2.6) | 40(10.3) | 10.838(0.001) |
| | No | 189(97.4) | 348(89.7) | |
| TOLAC | Yes | 24(12.4) | 44(11.3) | 0.866(0.386) |
| | No | 170(87.6) | 344(88.7) | |
| Mode of delivery (191; 3died before delivery) | SVD | 4(2.1) | 233(60.1) | 552(0.001) |
| | C/S | NA | 149(38.4) | |
| | Instrumental delivery | 3(1.5) | 6(1.5) | |
| | Vaginal breech delivery | 1(0.5) | 0 | |
| | Laparotomy | 183(94.3) | NA | |
| Cause of uterine rupture | Obstructed labor | 153(78.9) | NA | |
| | Scar dehiscence | 28(14.4) | NA | |
| | Use of Uterotonic drugs | 4(2.1) | NA | |
| | Use of instrumental delivery | 3(1.5) | NA | |
| | Other # | 6(3.1) | NA | |

# = Two after fundal pressure and four due to precipitated labor.

➤ TOLAC: trial of labor after cesarean section.

➤ SVD: spontaneous vagina delivery.

➤ C/S: cesarean section.

This study also showed the trend in the five-year study period. The trend of uterine rupture has decreased. This could be because of an increase in ANC follow-up, hospital delivery, or an increase in the deployment of gynecologists and integrated emergency obstetrics and surgery officers in the catchment area who can manage uterine rupture. Unlike the study done in Amhara region referral Hospitals [17]; this study showed a drop in the prevalence of uterine rupture in the five consecutive years. This difference could be because this research involves only one hospital and cases could have been managed in other Hospitals in the catchment area.

Many studies found a significant association between rural residency, age of the mother, TOLAC, trial of instrumental delivery, and uterine rupture. Studies done at Adma Hospitals [20], Dessie Hospital [18], Amhara referral Hospitals [17], and studies done in Norway [25] showed a significant association between TOLAC, and the age of the mother, residency, and uterine rupture. But this study didn't show a significant association between these variables and uterine rupture.

Many types of research showed ANC follow-up decreases maternal mortality and morbidity. ANC allows promoting skilled attendance at birth which in turn decreases intrapartum

**Table 3. Health system-related characteristics of mothers who gave birth at HUCSH 2020.**

| Variable | Response | Case (194) | Control (388) | X$^2$ (p value) |
|---|---|---|---|---|
| | | Frequency (%) | Frequency (%) | |
| Type of referring institution | None (from Home) | 1(0.5) | 152(39.2) | 218(0.00) |
| | Health center | 37(19.1) | 157(40.5) | |
| | Primary Hospital | 47(24.2) | 41(10.6) | |
| | General Hospital | 46(23.7) | 19(4.9) | |
| | Referral Hospital | 63(32.5) | 19(4.9) | |
| Number of health facilities before arrival at HUCSH | None (from Home) | 1(0.5) | 155(39.9) | 349. 758 (0.00) |
| | One | 33(17) | 206(53.1) | |
| | Two | 124(63.9) | 27(7) | |
| | Three and above | 36(18.6) | 0 | |
| Place of uterine rupture | Home | 51(26.3) | NA | 582 (0.00) |
| | Health Center | 90(46.4) | NA | |
| | Primary Hospital | 16(8.2) | NA | |
| | General Hospital | 18(9.3) | NA | |
| | Referral Hospital | 7(3.6) | NA | |
| | HUCSH | 3(1.5) | NA | |
| | During transport | 9(4.6) | NA | |
| The duration between admission and surgery (in hours) | Median | 1 | NA | |
| | Range | 1-7hr | NA | |
| Duration of hospital stay(days) | Median | 3 | 2.19(±2.3) | |
| | Range | 1–60 | 1–5 | |

complications like obstructed labor and uterine rupture. This study revealed that a lack of ANC follows up increases the likelihood of uterine rupture by more than seven times (AOR = 7.5; 95% CI: 1.9–30.3) as compared to those having four and above ANC follow-up. In line with this many studies showed a significant association between ANC follow-up and uterine rupture [11, 16, 19, 20]. The number of ANC follow up also has an association with uterine rupture. As compared to mothers who had four and above ANC follow-ups; those mothers who had less than four ANC follow up have an increased likelihood of uterine rupture by three times (AOR = 2.9; 95% CI: 1.3–6.5). The significance of ANC follow is obvious because every contact provider will have a chance to identify predisposing factors like malpresentation, fetal macrosomia, and history of the uterine scar. It also gives chance to council on the importance of hospital delivery and danger signs so that it decreases the occurrence of uterine rupture.

A number of gravidities are also an important factor associated with uterine rupture. The odds of having uterine rupture increase by more than three times for gravidity ≥5 as compared to gravidity of 1–4. This is supported by studies done in Norway [25], Pakistan [19], West Africa [20], and Ethiopia [11, 18, 20]. This could be because as parity increases the uterus may get less elastic and may predispose to uterine rupture. On the other hand, as parity increases the abdominal wall will be loose and this may predispose to fetal malpresentation which in return increases the chance of uterine rupture.

In developing countries, it is not uncommon to have the diagnosis of obstructed labor. In a study done at Jimma University, Ethiopia; obstructed labor is the commonest cause of uterine rupture [5]. In our study, there was a statistically strong association between obstructed labor and uterine rupture. Mothers who developed obstructed labor are more than 38—times more likely to develop uterine rupture than those who had no obstructed labor (AOR = 38.3; 95%

**Table 4. Outcome-related characteristics of mothers who gave birth at HUCSH 2020.**

| Variable | Response | Case (194) | Control (388) | X² (p value) |
|---|---|---|---|---|
| | | Frequency (%) | Frequency (%) | |
| Maternal condition on admission | Stable (normal BP) | 66(34) | 387(99.7) | 144.728(0.001) |
| | Shock | 128(66) | 1(0.3) | |
| Weight of the baby (in grams) | Mean (-+SD) | 3725.65(570.8) | 3202.6(+-551) | 108.59(0.001) |
| Status of the baby at birth | Alive with a good APGAR score | 5(2.6) | 367(94.6) | 500.223(0.001) |
| | Alive with a low APGAR score | 4(2.1) | 10(2.6) | |
| | Dead | 185(95.4) | 11(2.8) | |
| Condition of the mother on discharge | Alive | 183(94.3) | 388(100) | 22.424(0.001) |
| | Referred to another facility | 0 | 0 | |
| | Dead | 11(5.7) | 0 | |
| Type of uterine rupture (4 missing checks) | LUST (lower uterine segment transverse cesarean section) | 150(77.3) | NA | |
| | LUS vertical | 27(13.9) | NA | |
| | Upper uterine segment | 1(0.5) | NA | |
| | Posterior uterine rupture | 12(6.2) | NA | |
| | Died before surgery | 4(2.1) | NA | |
| Intra-operative complication | Bladder rupture | 8(4.1) | NA | |
| | Uterine artery involvement | 39(20.1) | NA | |
| Management (there is a missing 2) | Repair | 17(8.8) | NA | |
| | TAH (total abdominal hysterectomy) | 148(76.3) | NA | |
| | STH (subtotal hysterectomy) | 17(8.8) | NA | |
| | TAH and bladder repair | 8(4.1) | NA | |
| | Died before surgery | 4(2.1) | NA | |
| Post OP complication | Anemia that requires transfusion | 159(82) | NA | |
| | Vesicovaginal fistula | 1(0.5) | NA | |
| | Ureteric injury | 1(0.5) | NA | |

CI: 17.8–82.4). This is supported by research done in other parts of Ethiopia [17–19]. This is because, if the obstruction is not managed early, the distended uterine segment will rupture; especially in multiparous women.

This study also found a significant association between uterine rupture and fetal weight. The odds of developing uterine rupture increase by more than eight times (AOR = 8.4; 95% CI: 3.4–20.8). this is in line with the research done in Amhara Region referral Hospitals, North Ethiopia [17]. Fetal macrosomia can lead to fetal-pelvic disproportion which could lead to obstructed labor and then uterine rupture.

This research showed that mothers who had additional medical or obstetric disorders like cardiac disease, Hypertensive disorder, PROM, and APH, are less likely to develop uterine rupture than mothers without additional medical or obstetric disorders. Mothers without additional medical or obstetric conditions are more likely (AOR = 5.1; 95% CI: 1.6–16.1) to develop uterine rupture than mothers with additional medical or obstetric conditions. This could be because mothers with additional medical or obstetric disorders will visit the hospital early so that they will have better follow-up and management.

## Limitation of the study

Since the study is retrospective and data is collected from patient charts and logbooks, some important determinants of uterine rupture are missing. Moreover, the study is institution-based, it does not show the real magnitude of the population.

**Table 5. Determinants of uterine rupture of mothers who gave birth at HUCSH 2020.**

| Variables | Response | Case Frequency (%) | Controls Frequency (%) | COR (95% CI) | AOR (95% CI) | P-value |
|---|---|---|---|---|---|---|
| Residency | Urban | 46(13.2) | 303(86.8) | 1 | 1 | |
| | Rural | 148(63.5) | 85(36.5) | 11.47(7.6–17.3) | 1.95(0.9 5–4) | 0.07 |
| Age (years) | <35 | 148(30.3) | 341(69.7) | 1 | 1 | |
| | > = 35 | 46(49.5) | 47(50.5) | 2.26(1.44–3.54) | 0.63(0.23–1.87) | 0.36 |
| Number of ANC | 0 | 31(73.8) | 11(26.2) | 38.39(16.3–90.28) | 7.5(1.9–30.3) | **0.004** |
| | 1–3 | 147(48) | 159(52) | 12.59(7.23–21.94) | 2.9(1.3–6.5) | **0.01** |
| | > = 4 | 16(6.8) | 218(93.2) | 1 | 1 | |
| Gravidity | 1–4 | 95(21.2) | 353(78.8) | 1 | 1 | |
| | > = 5 | 99(73.9) | 35(26.1) | 10.51(6.72–16.44) | 3.4(1.4–8.3) | **0.008** |
| Gestational Age | <37 week | 3(15) | 17(85) | 0.343(0.099–1.18) | 2.8(0.56–14.3) | 0.2 |
| | > = 37week | 191(34) | 371(66) | 1 | 1 | |
| Onset of labor | Spontaneous | 187(36.1) | 331(63.9) | 1 | 1 | - |
| | Induced | 5(15.6) | 27(84.4) | 0.328(0.12–0.87) | 3.52(0.18–70.3) | 0.41 |
| Obstructed labor | Yes | 154(92.8) | 12(7.2) | 120.63(61.6–236.2) | 38.3(17.8–82.4) | **.000** |
| | No | 40(9.6) | 376(90.4) | 1 | 1 | |
| Use of uterotonics | Yes | 5(11.1) | 40(88.9) | 4.35(1.69–11.19) | 3.3(0.19–58.5) | |
| | No | 189(35.2) | 348(64.8) | 1 | 1 | 0.41 |
| Fetal weight | <4000 | 116(24) | 368(76) | 1 | 1 | 1 |
| | > = 4000 | 75(78.9) | 20(21.1) | 11.90(6.96–20.32) | 8.4(3.4–20.8) | **.0001** |
| Other medical or obstetric disorder | Yes | 9(12) | 66(88) | 1 | 1 | |
| | No | 185(36.5) | 322(63.5) | 4.2(2.1–8.65) | 5.1(1.6–16.1) | **0.005** |

## Conclusion and recommendation

The study showed a high prevalence of uterine rupture in the study area. It was also found that a decrease in the frequency of ANC follow-up, gravidity of ≥5, obstructed labor, and fetal weight of ≥4kg are significantly associated with uterine rupture as well as the presence of other medical or obstetric conditions is negatively associated with uterine rupture. So that not only the number but also the quality of ANC should be improved. The quality of intra-partal care, like the following labor with partograph, ultrasound estimation of fetal weight, and availability of cesarean section set up, should be improved. It is also recommended to encourage using take Long-acting contraceptive methods for grand multiparous women.

## Supporting information

**S1 Data.**
(SAV)

## Author Contributions

**Conceptualization:** Getnet Feleke, Dereje Zewdu, Abel Gedefawu, Muluken Gunta.

**Data curation:** Getnet Feleke, Abel Gedefawu, Muluken Gunta.

**Formal analysis:** Getnet Feleke, Abel Gedefawu, Muluken Gunta.

**Investigation:** Getnet Feleke, Temesgen Tantu, Abel Gedefawu, Muluken Gunta.

**Methodology:** Temesgen Tantu.

**Resources:** Dereje Zewdu.

**Software:** Dereje Zewdu, Mekete Wondosen.

**Supervision:** Dereje Zewdu, Mekete Wondosen.

**Validation:** Temesgen Tantu, Mekete Wondosen.

**Writing – review & editing:** Temesgen Tantu.

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
