## [Decision Letter · Decision Letter 0]

11 Nov 2022

PONE-D-22-20410Case-control study on determinants of uterine rupture among mothers who gave birth at Hawassa university comprehensive specialized hospital.PLOS ONE

Dear Dr. Tantu,

Thank you for submitting your manuscript to PLOS ONE. After careful consideration, we feel that it has merit but does not fully meet PLOS ONE’s publication criteria as it currently stands. Therefore, we invite you to submit a revised version of the manuscript that addresses the points raised during the review process.

We look forward to receiving your revised manuscript.

Kind regards,

Ibrahim Umar Garzali, MBBS, FWACS

Academic Editor

PLOS ONE

Journal Requirements:

2. We note you have included a table to which you do not refer in the text of your manuscript. Please ensure that you refer to Table 5 in your text; if accepted, production will need this reference to link the reader to the Table.

Reviewers' comments:

Reviewer's Responses to Questions

**Comments to the Author**

1. Is the manuscript technically sound, and do the data support the conclusions?

Reviewer #1: Yes

Reviewer #2: Partly

Reviewer #3: Partly

Reviewer #4: Yes

2. Has the statistical analysis been performed appropriately and rigorously? 

Reviewer #1: Yes

Reviewer #2: No

Reviewer #3: Yes

Reviewer #4: No

3. Have the authors made all data underlying the findings in their manuscript fully available?

Reviewer #1: No

Reviewer #2: Yes

Reviewer #3: Yes

Reviewer #4: Yes

4. Is the manuscript presented in an intelligible fashion and written in standard English?

Reviewer #1: Yes

Reviewer #2: Yes

Reviewer #3: Yes

Reviewer #4: No

5. Review Comments to the Author

Reviewer #1: The manuscript covers a very important aspect in obstetric practice and it is written well. Although it does not add much to the existing literature relating to the understanding of the uterine rupture, it reviews the risk factors associated with it.

Reviewer #2: 1. As stated in the Background section, even in Ethiopia the uterine rupture due to prior uterine scar was less than other countries, however, this is still one of the key issues that couldn't be omitted. In the current study, the cases included the prior uterine scar cases were enrolled and focused only at the previous Cesarean section, but no further investigation was available.

2. The listed variables in the manuscript might be independent, that means the statistical confounding effect among them should be taken into consideration.

3. The definitions of uterine rupture, as mentioned in the Background section, were categorized by fully tears and partial tears. What are the criteria to confirm the situation, especially the partial rupture, by ultrasound or any other pathologic proof?

4. There are too many abbreviations that need to be clarified, for example, ANC, PROM, TOLAC and so on.

Reviewer #3: FULL REVIEW IS IN THE ATTACHED DOCUMENT

The manuscript is well-thought out and addresses an issue of public health importance.

Test statistic estimates of differences between the cases and the control will be relevant to answer some of the study objectives. The presentation of both the crude odds ratio and the adjusted odds ratio, with the p-value is excellent in order to know the accuracy and precision of study estimates.

The discussion was intelligently done by giving us the study finding, comparing it with other studies, informing us of the reasons for the variations and similarities and its implication to public health and obstetric practice, especially on low- and middle income countries.

Overall, the manuscript is scientifically and analytically sound; albeit with minor revisions and corrections in some areas. It is a worthy addition to the current pool of knowledge on uterine rupture in low- and middle income countries.

ADEOYE, Philip Adewale

Jos University Teaching Hospital, NIGERIA.

Reviewer #4: I want to thank the editor-in-chief for the opportunity to review the manuscript. I have a few comments and suggestions listed below and attached the CASP checklist for further modification by the authors. Although the paper is a case-control study for identifying the determinants of uterine rupture, the design can not answer the problem's prevalence because it is a hospital-based study. In the abstract, the confidence interval should be listed after the effect size of the factors. The authors are advised to review and edit their paper for clarity and to use future tenses in the methodology. Also, the identified factors were not new; what would the study add? In table 4, the authors used abbreviations but did not disclose them in the legends. As some of these terms are not well known in the scientific community, they must have a legend below the table. Please refer to the CASP checklist for further corrections and modifications.

6. PLOS authors have the option to publish the peer review history of their article (what does this mean?). If published, this will include your full peer review and any attached files.

Reviewer #1: **Yes: **Nourah Al Qahtani

Reviewer #2: No

Reviewer #3: **Yes: **ADEOYE, Philip Adewale

Reviewer #4: No

---

## [Author Response · Author response to Decision Letter 0]

18 Nov 2022

Response to reviewers:

I am delighted to have my research article get consideration for publication in this renowned journal. I would like to say thank you for your time to evaluate our paper. All comments and suggestions were accepted and corrected properly and then attached. Here are my responses individually

Reviewer #1: The manuscript covers a very important aspect in obstetric practice and it is written well. Although it does not add much to the existing literature relating to the understanding of the uterine rupture, it reviews the risk factors associated with it

• Thank you for your excellent and constructive comments, we accepted all comments and tried to add evidences for existing knowledge.

Reviewer #2: 

• As stated in the Background section, even in Ethiopia the uterine rupture due to prior uterine scar was less than other countries, however, this is still one of the key issues that couldn't be omitted. In the current study, the cases included the prior uterine scar cases were enrolled and focused only at the previous Cesarean section, but no further investigation was available.

Response: I am very sorry for not making it clear to the reviewer and reader, the comment was accepted but unfortunately, we had uterine scars done only for cesarian section no cases with scars done for other uterine pathology. Additionally, we could not calculate the contribution of uterine scar dehiscence for uterine rupture since we don’t have the total number of pregnant ladies who had previous cesarean section.

The listed variables in the manuscript might be independent, that means the statistical confounding effect among them should be taken into consideration.

• Response: Sorry for not making it clear for the reviewer. The comment accepted and corrected. Those independent variables were analyzed with bivariate regression analysis then the variables with strong association with uterine rupture were taken to multivariate analysis and multicollinearity tests done by taking variance inflator factor VIF > 10 to control the confounding variables. So, only variables with strong association during multivariate analysis were taken as factors

The definitions of uterine rupture, as mentioned in the Background section, were categorized by fully tears and partial tears. What are the criteria to confirm the situation, especially the partial rupture, by ultrasound or any other pathologic proof?

• Response: I would like to say thank you for your time and courage to have comment. As we mentioned in the background section, the incomplete rupture is when the outer layer of the uterus is intact while other layers are torn out. Commonly, it is intraoperative diagnosis by the surgeon. However, it can be diagnosed incidentally by ultrasound if it occurs spontaneously before the onset of labor or when there is strong suspicion by the managing physician during labor.

There are too many abbreviations that need to be clarified, for example, ANC, PROM, TOLAC and so on

• We are very Sorry for not making it clear for the reviewer. The comment accepted and corrected

Reviewer #3

The manuscript is well-thought out and addresses an issue of public health importance.

Test statistic estimates of differences between the cases and the control will be relevant to answer some of the study objectives. The presentation of both the crude odds ratio and the adjusted odds ratio, with the p-value is excellent in order to know the accuracy and precision of study estimates.

The discussion was intelligently done by giving us the study finding, comparing it with other studies, informing us of the reasons for the variations and similarities and its implication to public health and obstetric practice, especially on low- and middle income countries.

Overall, the manuscript is scientifically and analytically sound; albeit with minor revisions and corrections in some areas. It is a worthy addition to the current pool of knowledge on uterine rupture in low- and middle income countries

• Thank you for your excellent and constructive comments, we accepted all comments and corrected the comments. Line by line response as follows.

Abstract: Confidence intervals not included with the odd ratios

• We are very sorry for not including the confidence intervals, all comments are accepted and corrected. 

Introduction: Objective looks like a justification.

• We are very sorry for the inconvenience we created, all comments are accepted and corrected

Sampling frame may not be necessary if probability sampling method is not used.

• Sorry for creating this confusion. The comment is accepted and corrected. We did not used the probability sampling.

“…tasted” seems to be a typographical error

• Thank you for your comment. It is accepted and corrected

The prevalence estimates of 1.09% description not sufficient. Adding 95%CI to this prevalence may be better?

• Thank you for your comment. It is accepted and corrected

Test statistics not reported in comparing the differences between the cases and controls in Tables 1,2,3and 4

• Sorry for our failure to include the comparison. All the comments and suggestions are accepted and corrected.

Total number eventually studied for each variable are not reported to determine missing data.

• Sorry for our failure. All the comments and suggestions are accepted and corrected.

Range may not be necessary to be reported. (Table 1 - under age) 

• Sorry for our failure. All the comments and suggestions are accepted and corrected

Meaning of TOLAC may not be clear to a lay- reader (Table 2)

• I am very sorry for not making it clear to the reviewer and reader, the comments are accepted and corrected.

Since this observation appears not normally distributed, mean (SD) should not be reported. I will suggest that Median (interquartile range) be reported.

• I am very sorry for not making it clear to the reviewer and reader, the comments are accepted and corrected.

Since this Observation appears not normally distributed, mean (SD) should not be reported. (Table 3) under “duration between admission and surgery” and “Duration of hospital stay”

• Sorry for creating this confusion. The comment is accepted and corrected

Range may not be necessary since the observation appears normally distributed. (Table 4- under “weight if the baby”

• Sorry once again, the comments are accepted and corrected.

The meaning of “LUST”, “TAH” and “STH” may not be clear to a lay- reader. (Table 4)

• I am very sorry for not making it clear to the reviewer and reader, the comments are accepted and corrected

“Having ANC follow up” seems not intended. It looks like you meant “Having no ANC follow-up?” check this up by comparing with your table and make necessary correction.

• Sorry for the mistake, the comments are accepted and corrected.

The Reported AOR and their 95%CI seems not corresponding with them results shown on the Table 5 for “Number of ANC follow up (less than four ANC)”; “gravidity more than 5” and “fetal weight greater than 4000 grams”

• Sorry once again, the comments are accepted and corrected.

Paragraph 1 and 2, paragraph 3 and 4 though discussing the same point are separate. the two paragraphs can be merged since they are focused on a particular result.

• Thank you for your constructive comments. All the comments are accepted and corrected.

Reference 8 appears not properly done

• Thank you for your time and courage to have such constructive comments. The comments are accepted and corrected. Reff no 8 removed from lists.

Reviewer #4

 I want to thank the editor-in-chief for the opportunity to review the manuscript. I have a few comments and suggestions listed below and attached the CASP checklist for further modification by the authors. Although the paper is a case-control study for identifying the determinants of uterine rupture, the design cannot answer the problem's prevalence because it is a hospital-based study. In the abstract, the confidence interval should be listed after the effect size of the factors. The authors are advised to review and edit their paper for clarity and to use future tenses in the methodology. Also, the identified factors were not new; what would the study add? In table 4, the authors used abbreviations but did not disclose them in the legends. As some of these terms are not well known in the scientific community, they must have a legend below the table. Please refer to the CASP checklist for further corrections and modifications.

• Thank you for your time and willingness to evaluate the manuscript. All the comments and suggestion are accepted and then corrected. Specific line by line responses for the comments are as follows.

• As it is suggested by the reviewer, it would be not possible to do prevalence in case control study. However, we tried to get the whole delivery lists in the past five years then we calculated the prevalence and trends of the rupture to show the pattern in the past five years which helps us to have better understand of the problem. Otherwise, it is correct and acceptable comment.

• Thank you for your time and effort to evaluate the paper. We used the past sentences in the methodology since the data collection and the study has been already done. 

• The study further strengthens the existing knowledge on factors associated with uterine rupture and the new thing may be the having ANC follow up less than 4 is also associated with uterine rupture. Not only the presence or absence of ANC follow up but also quality of ANC is important.

• Sorry for not making clear for reader and reviewer. All the abbreviation are corrected.

From CASP check list:

The paper did not describe how they selected 194 cases from 247 total uterine rupture.

• Sorry once again for not making clear for the reviewer and reader. Total cases of uterine rupture are 247 cases but when we exclude the charts with medico legal case, lost charts and those chats with inadequate information, total of 200 charts become eligible for the study so we took 194 chats for 200 chats. We removed 6 charts randomly.

However, prevalence of uterine rupture cannot be answered through hospital-based study.

• As it is suggested by the reviewer, it would be not possible to do prevalence in case control study. However, we tried to get the total delivery lists in the past five years then we calculated the prevalence and trends of the rupture to show the pattern in the past five years which helps us to have better understand of the problem. Otherwise, it is correct and acceptable comment.

Please perform sensitivity analysis:

• Sorry for not making it clear for the reader and reviewer. The comments and suggestions are accepted and corrected. We did multivariate analysis to control the confounding factors for those variables which had association during bivariate analysis. Additionally, the multicollinearity tests performed by taking variance inflation factor (VIF) > 10

The effect was not precise and the interval was very hide, suggesting unidentified confounders

• Humble apology for the all drawbacks of the study. Sine it is retrospective chart review it has some drawbacks mentioned at limitation of the study.

---

## [Editor Report · Decision Letter 1]

2 Dec 2022

Case-control study on determinants of uterine rupture among mothers who gave birth at Hawassa university comprehensive specialized hospital.

PONE-D-22-20410R1

Dear Dr. Tantu,

We’re pleased to inform you that your manuscript has been judged scientifically suitable for publication and will be formally accepted for publication once it meets all outstanding technical requirements.

Kind regards,

Ibrahim Umar Garzali, MBBS, FWACS

Academic Editor

PLOS ONE
---

## [Editor Report · Acceptance letter]

4 Jan 2023

PONE-D-22-20410R1 

Case-control study on determinants of uterine rupture among mothers who gave birth at Hawassa university comprehensive specialized hospital. 

Dear Dr. Tantu:

I'm pleased to inform you that your manuscript has been deemed suitable for publication in PLOS ONE. Congratulations! Your manuscript is now with our production department. 

Kind regards, 

on behalf of

Dr. Ibrahim Umar Garzali 

Academic Editor

PLOS ONE